# Integration of a multicomponent intervention for hypertension into primary healthcare services in Singapore—A cluster randomized controlled trial

**Tazeen Hasan Jafar**[1,2,3]*, **Ngiap Chuan Tan**[4], **Rupesh Madhukar Shirore**[1], **John Carson Allen**[5], **Eric Andrew Finkelstein**[1], **Siew Wai Hwang**[4], **Agnes Ying Leng Koong**[4†], **Peter Kirm Seng Moey**[4], **Gary Chun-Yun Kang**[4], **Chris Wan Teng Goh**[4], **Reena Chandhini Subramanian**[4], **Anandan Gerard Thiagarajah**[6], **Chandrika Ramakrishnan**[1], **Ching Wee Lim**[1], **Jianying Liu**[4], for SingHypertension Study Group[¶]

1 Program in Health Services & Systems Research, Duke-NUS Medical School, Singapore, 2 Department of Renal Medicine, Singapore General Hospital, Singapore, 3 Duke Global Health Institute, Durham, North Carolina, United States of America, 4 SingHealth Polyclinics, Singapore, 5 Center for Quantitative Medicine, Office of Research, Duke-NUS Medical School, Singapore, 6 National University Polyclinics, Singapore

† Deceased.
¶ Members of SingHypertension Study Group are listed in the Acknowledgments and Section S1 in S1 Appendix.
* tazeen.jafar@duke-nus.edu.sg

**Data Availability Statement:** Data are available, with some restrictions due to the study's ethics approval. Interested individuals can contact

## Abstract

### Background

Despite availability of clinical practice guidelines for hypertension management, blood pressure (BP) control remains sub-optimal (<30%) even in high-income countries. This study aims to assess the effectiveness of a potentially scalable multicomponent intervention integrated into primary care system compared to usual care on BP control.

### Methods and findings

A cluster-randomized controlled trial was conducted in 8 government clinics in Singapore. The trial enrolled 916 patients aged ≥40 years with uncontrolled hypertension (systolic BP (SBP) ≥140 mmHg or diastolic BP (DBP) ≥90 mmHg).

Multicomponent intervention consisted of physician training in risk-based treatment of hypertension, subsidized losartan-HCTZ single-pill combination (SPC) medications, nurse training in motivational conversations (MCs), and telephone follow-ups. Usual care (controls) comprised of routine care in the clinics, no MC or telephone follow-ups, and no subsidy on SPCs. The primary outcome was mean SBP at 24 months' post-baseline. Four clinics (447 patients) were randomized to intervention and 4 (469) to usual care. Patient enrolment commenced in January 2017, and follow-up was during December 2018 to September 2020. Analysis used intention-to-treat principles. The primary outcome was SBP at 24 months. BP at baseline, 12 and 24 months was modeled at the patient level in a likelihood-

SingHealth institutional review board at <irb@singhealth.com.sg>.

**Funding:** The trial was funded via Senior Clinical Scientist award from the National Medical Research Council, Singapore (CSA-SI/0005/2015) (THJ) https://www.nmrc.gov.sg/grants. The funder of the study had no role in study design, data collection, data analysis, decision to publish, or preparation of the manuscript.

**Competing interests:** The authors have declared that no competing interests exist. Author Agnes Koong was unable to confirm their authorship contributions. On their behalf, the corresponding author has reported their contributions to the best of their knowledge.

**Abbreviations:** ACEI, angiotensin-converting enzyme inhibitor; ACR, albumin to creatinine ratio; ARB, angiotensin II receptor blocker; BMI, body mass index; BP, blood pressure; CKD, chronic kidney disease; COVID-19, Coronavirus Disease 2019; CRC, clinical research coordinator; CVD, cardiovascular disease; DBP, diastolic BP; DDF, denominator degrees of freedom; eGFR, estimated glomerular filtration rate; FBG, fasting blood glucose; FRS, Framingham risk score; ITT, intention-to-treat; LDL, low-density lipoprotein; MMRM, mixed model repeated measure; MRC, Medical Research Council; RAAS, renin-angiotensin-aldosterone system; RCT, randomized controlled trial; SAE, serious adverse event; SAP, statistical analysis plan; SBP, systolic BP; SPC, single-pill combination; VAS, visual analogue scale.

based, linear mixed model repeated measures analysis with treatment group, follow-up, treatment group × follow-up interaction as fixed effects, and random cluster (clinic) effects.

A total of 766 (83.6%) patients completed 2-year follow-up. A total of 63 (14.1%) and 87 (18.6%) patients in intervention and in usual care, respectively, were lost to follow-up. At 24 months, the adjusted mean SBP was significantly lower in the intervention group compared to usual care (−3.3 mmHg; 95% CI: −6.34, −0.32; $p = 0.03$). The intervention led to higher BP control (odds ratio 1.51; 95% CI: 1.10, 2.09; $p = 0.01$), lower odds of high (>20%) 10-year cardiovascular risk score (OR 0.67; 95% CI: 0.47, 0.97; $p = 0.03$), and lower mean log albuminuria (−0.22; 95% CI: −0.41, −0.02; $p = 0.03$). Mean DBP, mortality rates, and serious adverse events including hospitalizations were not different between groups. The main limitation was no masking in the trial.

## Conclusions

A multicomponent intervention consisting of physicians trained in risk-based treatment, subsidized SPC medications, nurse-delivered motivational conversation, and telephone follow-ups improved BP control and lowered cardiovascular risk. Wide-scale implementation of a multicomponent intervention such as the one in our trial is likely to reduce hypertension-related morbidity and mortality globally.

## Trial registration

Trial Registration: Clinicaltrials.gov NCT02972619.

## Author summary

### Why was this study done?

- High blood pressure (BP) continues to be the leading risk factor for mortality globally.
- BP control remains suboptimal even in high-income countries.
- Evidence is scarce on rapidly scalable health systems strategies to control BP.
- Previous studies are of short duration (majority <6 months) with mixed results, with some showing benefit and the others neutral.

### What did the researchers do and find?

- We conducted a pragmatic, cluster trial in clinics in Singapore to evaluate a multicomponent intervention comprised of training physicians and nurses, motivational conversation, telephone follow-ups, and subsidies on single-pill combination (SPC) antihypertensive medications.
- We randomized 4 clinics to the multicomponent intervention and 4 to usual care (controls) and recruited 916 patients aged 40 years or older with uncontrolled hypertension.

- Over 24 months of follow-up, the intervention led to a significant lowering of systolic BP (SBP), and improvement in BP control, use of antihypertensive medications, cardiovascular risk and albuminuria, and was not associated with any safety concerns.

### What do these findings mean?

- A multicomponent intervention that leverages the primary care infrastructure and addresses barriers to hypertension care at multiple (patient, physician, health systems) levels could lead to an improvement in BP control and lower cardiovascular risk.

- Given the potential ease of adaptation into both public and private health systems and rapid scalability, high priority implementation of a multicomponent intervention program such as the one in our trial would represent a significant step forward toward reducing hypertension-related morbidity and mortality globally.

## Introduction

High blood pressure (BP) is the leading risk factor for mortality globally, accounting for 19% (10.8 million) of all deaths [1]. Despite numerous clinical practice guidelines for hypertension management, implementation remains inadequate with BP control less than 30% even in high-income countries [2,3].

Several barriers to hypertension care have been identified at multiple levels worldwide [4,5]. The significant patient-level challenges include **health illiteracy** about hypertension and related cardiovascular disease (CVD), lack of motivation to adopt a healthy lifestyle, poor medication adherence, concern about side effects from treatment, complex dosing regimens, and high cost [6,7]. The physician-level issues are clinical inertia to initiate or intensify treatment, reluctance to prescribe single-pill combination (SPC) antihypertensive drugs, which are tolerated better than individual drugs and improve adherence, lack of training in health communication [8,9]. The health systems-level barriers are inadequate funding, lack of performance standards, and time constraints in busy clinical practices [6,7,10].

One strategy to improve hypertension management involves adopting multicomponent intervention to address many of the above barriers yet are scalable in the public sector [11]. However, the vast majority of studies testing the effectiveness of multicomponent interventions are of short duration (6 months or less), and did not address the barriers at all levels [11]. The few studies with longer-term follow-up from did not show benefit persisting beyond 1 year [12,13].

Thus, a scarcity of information exists on effective, scalable, and potentially sustainable primary care interventions for managing hypertension in high-income countries.

By leveraging the public sector primary care clinic infrastructure, we conducted a pragmatic, cluster-randomized controlled trial (RCT) over 2 years in Singapore to evaluate the effectiveness of a potentially scalable, multicomponent intervention designed specifically to manage hypertension.

We adapted the intervention strategy from our pilot feasibility study to fit the primary care clinic workflow, receiving favorable feedback from stakeholders including the healthcare providers and patients with hypertension [14,15]. The intervention addressed several of the

barriers identified above and included physician training on risk-based management, motivational conversations with patients by trained nurses, telephone follow-ups of all patients with hypertension by trained nurses, and subsidy on SPC antihypertensive medication.

The comparator was usual care in the primary care clinics. We hypothesized that a multicomponent intervention tailored to the existing primary care infrastructure would be more effective than usual care in lowering systolic BP (SBP) among adults with uncontrolled hypertension.

## Methods

### Design and oversight

The study was a pragmatic cluster RCT in 8 government primary care clinics in Singapore conducted according to the framework of the United Kingdom Medical Research Council (MRC) for implementing complex intervention trials [16,17].

Because intervention delivery was via the primary care system, a cluster RCT design was chosen to minimize intervention "spill-over"—participants in the usual care arm receiving any component of the multicomponent intervention.

### Randomization and masking

We used computer-generated codes to randomly assign 4 clinics to the multicomponent intervention and 4 to usual care. Following randomization, 916 individuals with uncontrolled hypertension were recruited from these clinics. Patient enrollment started in January 2017 and ended in May 2018, and follow-up was conducted from December 2018 to September 2020. There was no masking in the trial. The trial protocol (S1 Protocol in S1 Appendix) and statistical analysis plan (SAP) (S1 Statistical Analysis Plan in S1 Appendix) were published previously [15,18].

The Ethics Review Committees at SingHealth institutional review board and National University of Singapore approved the study. Informed consent was signed by all participants before screening. An independent Data Safety and Monitoring Committee reviewed the trial conduct. Funders had no role in the design, conduct, analysis, interpretation, or reporting of results.

### Study participants

Public sector clinics in Singapore provide subsidized primary care services to a multiethnic patient population from diverse socioeconomic backgrounds [19]. These clinics are staffed by physicians and nurses, and have onsite pharmacies and laboratories. Patients are assigned to a pool or team of physicians at every visit for a nominal co-payment. Patients may also choose to be seen by a family physician in the polyclinic albeit at a significantly higher (2 to 3 times) cost. Each provider attends to about 50 to 70 patients daily, with an average encounter time of about 7 to 10 minutes per patient [20]. Medications for chronic conditions are dispensed on prescriptions only. All major classes of antihypertensive medications (thiazide-like diuretics, calcium channel blockers, renin-angiotensin-aldosterone-system blockers, and beta blockers) are available at the pharmacies, and their generic formulation are subsidized. Generic and branded SPC antihypertensive medications are also available, albeit neither are subsidized as part of usual care.

We enrolled patients aged ≥40 years with previously diagnosed hypertension (SBP ≥140 mmHg or diastolic BP (DBP) ≥90 mmHg) based on mean of last 2 of 3 measurements who had visited an enrolled clinic at least twice during the prior 12 months and were Singaporean citizens or permanent residents.

## Procedures

The multicomponent intervention is described in detail in the protocol paper [15]. Briefly, it incorporated the following components:

1. Physician training on risk-based management: Physicians were trained in a risk-based approach treatment algorithm requiring cardiovascular risk stratification of patients based on the 10-year Framingham risk score (FRS) for CVDs modified for the local population that is already embedded in the electronic health record system at the clinics [21]. SPC recommended for patients at high-risk, i.e., those with high ($\geq$20%) 10-year CVD risk based on FRS, or those with preexisting CVD, or chronic kidney disease (CKD). Statins were also initiated if low-density lipoproteins (LDL) cholesterol was 2.6 mmol/L (100 mg/dl) or higher, as per the algorithm [22]. The least expensive, generic, SPC angiotensin II receptor blocker (ARB) and a thiazide diuretic (losartan-HCTZ) available in the polyclinic pharmacies was preferred. For patients at medium or low risk, the first-line antihypertensive medication was calcium channel blockers, followed by angiotensin-converting enzyme inhibitors (ACEIs) (preferably in those aged <55 years) or thiazide diuretics (preferably in those $\geq$55 years) initiated at half-standard dose and up-titrated as necessary. Use of beta blockers as first-line antihypertensive agents without a compelling indication was discouraged. The general target BP was <140/90 mmHg. However, for hypertensive individuals with proteinuria or preexisting CVD, the target BP was <130/80 mmHg. Physicians completed a standardized management checklist. The initial training session was over 2 hours, with re-training at 3 months and then 12 and 24 months. Details of the checklists and algorithm are described in Sections S2 to S5 in S1 Appendix.

2. Motivational conversations: A motivational conversation curriculum for hypertensive patients was developed in a feasibility study [23]. The counseling approach is intended to empower patients with hypertension to manage risk factors, set self-care goals for a healthy lifestyle (diet, physical activity, alcohol intake moderation, and smoking cessation), use a home BP monitor, and adhere to medication regimens [24]. A consultant specialist psychologist trained the clinic nurses in motivational conversation skills for 4 hours using the curriculum [23,24]. Refresher training was provided at months 12 and 24. Trained nurses delivered one 15- to 20-minute face-to-face motivational conversation session to patients identified at high risk of CVD.

3. Telephone-based follow-up by trained nurses: The nurses trained in motivational conversations also followed all hypertensive patients by telephone for adherence reinforcement to healthy lifestyles and treatment, using a standardized checklist. The calls were scheduled at monthly intervals for the first 3 months and then 3-month intervals for the 2-year duration of the project for all patients. Regular meetings were scheduled at 6- to 8-week intervals among physicians and nurses to review patients not achieving BP targets to provide advice or action, as needed.

4. Subsidy on SPC medications: A 50% discount on losartan-HCTZ SPC medication was offered to patients at high risk of CVD in the intervention clinics throughout the trial. Discounts were channeled through clinic pharmacies. The savings amounted to about S$7 per month per patient.

Usual care consisted of existing clinic services with routine care by physicians and nurses. Physicians were free to prescribe medications of their choice. Although SPC medications (including losartan-HCTZ costing S$12 to 15, per 1-month supply) were available in the clinic pharmacies, there was no trial-based subsidy for usual care participants. However, all

participants in usual care received subsidies on single agent generic antihypertensive medications as per routine clinical practice.

## Baseline screening

An on-site trained clinical research coordinator (CRC) at the participating clinics approached potentially eligible patients with a prior diagnosis of hypertension and invited them for pre-screening. CRCs measured BP 3 times with 3-minute intervals between readings, with the patient in a sitting position, arm rested, using an upper arm calibrated automated Omron device (HEM-7130). Individuals with SBP $\geq$140 mmHg or DBP $\geq$90 mmHg (mean of last 2 of 3 BP measurements) were considered to have uncontrolled BP. Thus, the patients knew the group to which their practice was assigned at enrollment. Informed consent specific for intervention and usual care participants was obtained for further screening. Patients with active systemic illness (fever, known liver disease), clinically unstable heart failure, advanced CKD (estimated glomerular filtration rate (eGFR) <40 ml/minute/1.73m$^2$), proteinuria $\geq$3 g/d, hospitalization in the prior 4 months, deemed mentally unfit to give informed consent, and pregnant women, were excluded.

This trial is reported according to the Consolidated Standards of Reporting Trials (S1 CONSORT Checklist).

The baseline, 12- and 24-month questionnaire was administered to all eligible patients and information was collected on sociodemographics (baseline only), comorbidities, diet, lifestyle and tobacco use, direct and indirect healthcare cost, and quality of life (EuroQol-5 Dimension-5 Level questionnaire (EQ-5D-5L)).

Anthropometric measurements (weight, height, waist circumference) were obtained. Fasting blood samples for serum electrolytes, glucose, and lipids were collected as part of the hypertension panel test. A random urine sample was collected for albumin, creatinine, and sodium.

## Outcomes assessments

All participants were assessed in the clinic by trained CRC independent to treatment at 1-year and 2-year post-baseline.

Standardized measurement of BP was obtained using the same procedure as in baseline. CRCs also performed telephone follow-up assessments every 4 months over the 2-year study period.

Information on adverse events (including falls, hypotension, cough, hyperkalemia, musculoskeletal pain, peripheral edema, coronary heart disease, stroke, and heart failure) were reported. Hospitalizations and deaths were tracked (details are provided in the protocol). Deaths due to myocardial infarction, heart failure, or stroke (per ICD-10 codes) were categorized as cardiovascular deaths. Fasting blood and random urine samples were collected at 24 months. Patients were reimbursed for cost of travel to the clinic for their final follow-up visit.

For use in a future cost-effectiveness analysis, costs related to the training and time of nurses, physicians' training, telephone follow-ups, and SPC subsidy were recorded.

Reporting in this publication is consistent with the CONSORT statement.

## Outcomes

The primary endpoint was SBP based on a mean of last 2 of 3 measurements at the 2-year post-baseline final follow-up.

Prespecified key BP-related secondary outcomes included percentage of participants with BP control (SBP <140 mmHg and DBP <90 mmHg), DBP, number of antihypertensive

medications, high (>20%) 10-year CVD risk based on FRS [21], and albuminuria measured as urine albumin to creatinine ratio (ACR) at 2-year post-baseline. Other secondary outcomes also at 2-year post-baseline included various thresholds of BP control, lifestyle measures (physical activity, body mass index (BMI), waist circumference, smoking status, intake of fruits and vegetables, dietary sodium intake (urinary sodium excretion)), clinical measures and outcomes (lipid levels, eGFR, dose of antihypertensive medications) [25], integrated 10-year CVD risk score (validated for a Chinese population) [26], fasting blood glucose (FBG), new-onset diabetes, and participant-reported health status. The latter was measured according to the mean score on the visual analogue scale (VAS) of the EQ-5D-5L (range, 0 to 100, with higher scores indicating better health) and mean score on the EQ-5D-5L utility index calculated using the Indonesian value set (range, −0.865 to 1, with higher scores indicating better health) [27]. In addition, the safety events included mortality, any serious adverse event (SAE), SAEs of special interest (cardiovascular death, hospitalization for CVDs due to coronary heart disease or acute coronary syndrome or elective revascularization or myocardial infarction, stroke, heart failure, peripheral vascular disease, and peripheral edema), adverse events (AEs), and hospitalizations potentially related to the intervention.

## Statistical analysis

An a priori sample size of $n = 125$ patients was recruited at each of the 8 study clinics (4 intervention, 4 usual care) to detect a (cohen) effect size of 0.28 between the intervention and usual care at the 2-year follow up with 80% power at $\alpha = 0.05$ assuming a within-clinic intra-class correlation of 0.01, based on assumptions from previous trials, including our own [28,29].

An SAP was published previously [18]. The original analysis for the primary outcome for the trial was patient-level change from baseline in SBP at 24-month post randomization. However, at the publication of the SAP, the analysis was modified to SBP at 24 months in a clinic-level comparison. At that time, an unexpectedly higher potential for patient dropout was anticipated, as the Coronavirus Disease 2019 (COVID-19) pandemic led to delays in clinic attendance. The SAP reviewer underscored that analysis of change from baseline can be inefficient, leads to unnecessary exclusion of subjects with missing data, and that a superior method would be to analyze a multivariate (repeated) outcome that includes the baseline measure of the response [30,31]. Therefore, we elected to use a repeated measures analysis of SBP inclusive of baseline SBP as one of the outcomes vector in the model [18]. Notwithstanding the modification, the analysis on the original primary outcome of patient-level change from baseline in SBP is reported in a sensitivity analysis.

Briefly, all analyses were performed based on the intention-to-treat (ITT) principle. BP at baseline, 12 and 24 months was modeled at the participant level in a likelihood-based, linear mixed model repeated measures (MMRMs) analysis with cluster random effects for clinic assuming participant and cluster (clinic) Gaussian error distributions, identity link function, using Satterthwaite approximation denominator degrees of freedom (DDF). Treatment arm and follow-up time and treatment × follow-up time interaction were modeled as fixed effects and random effects for clusters (clinics) and participants. All other secondary outcomes were analyzed using the same modeling approach but incorporating generalized techniques appropriate for the type of outcome and using between-within DDF.

Sensitivity analyses: (1) ITT for outcome of change from baseline in SBP and key BP-related secondary outcomes at 24 months with baseline BP (or baseline level of the respective secondary outcome) as a covariate performed at the patient level using between-within DDF, per ITT (2) per protocol analysis of the primary outcome using MMRM. Participants who were high CVD risk as per the checklist and did not receive single pill combination (SPC) in the

multicomponent intervention (*n* = 73) as per protocol were excluded from this analysis. (3) Primary outcome restricted to patients completing the 24-month follow-up, (4) ITT for primary outcome using MMRM with adjustment for clinically important variables at baseline (including age, gender, waist circumference, diabetes, and 10-year FRS CVD risk score), (5) ITT after multiple imputations, in which missing values of 1- (15.0%) and 2-year (16.4%) SBP were replaced through a process of multiple imputation (details in Section S6 in S1 Appendix) [32]. After imputation, each of the 20 imputed datasets was analyzed using a MMRM model similar to the approach used for the non-imputed dataset. The parameter estimates obtained from each dataset were pooled for inference and (6) primary outcome using MMRM analysis restricted to participants followed up prior to 12 March 2020 (onset of COVID-19 pandemic). Details regarding sensitivity analyses are provided in the Supporting information (Section S6 in S1 Appendix).

Cost: The incremental cost of intervention delivery was computed. These included the cost of administration and oversight, physicians' and nurses' training, implementation of motivational conversation, subsidy for SPC antihypertensive medication, cost of other antihypertensive and lipid lowering medications, and cost of related laboratory (safety) tests. Details are reported in the Section S7 in S1 Appendix.

A detailed incremental cost-effectiveness analysis taking a lifetime perspective will be the subject of another paper.

## Results

A total of 3,836 individuals aged ≥40 years were prescreened at the clinics, and 916 (23.9%) with uncontrolled hypertension who met the eligibility criteria were enrolled (447 and 469 participants in intervention and usual care, respectively) from January 2017 to May 2018 (Fig 1).

The 24-month follow-up ended in September 2020 with 85.9% and 81.4% retention in the intervention and usual care arms, respectively (Fig 1).

Mean (SD) age of trial participants was 64.5 (9.8) years; 49.6% were women; 73.5% were Chinese, 13.3% Malay, 9.7% Indian, and 3.5% other ethnicities; 73.3% were overweight or obese (Asian cutoff of 23.5 kg/m$^2$ or more) [33]; 34.3% had diabetes; and 59.8% had at least 1 comorbid chronic disease (diabetes, CKD, self-reported heart disease, or stroke) (Table 1 and Table A of Section S8 in S1 Appendix).

By design, all participants had uncontrolled BP at baseline. The mean SBP (SD) and DBP (SD) were 149.3 (13.0) mmHg and 88.3 (9.8) mmHg, respectively. Baseline characteristics were generally balanced between the intervention and usual care arms (Table 1 and Table A of Section S8 in S1 Appendix).

The intervention had high implementation fidelity: 100% of the invited physicians in the intervention clinics were trained and completed the participant management checklist; 99.6% of participants received at least 1 telephone-based follow-up, and a mean (95% CI) of 7.2 (7.0, 7.4) calls per participant were delivered during 2 years; 94.5% of high-risk participants received 1 motivational conversation session; 63.7% received prescribed SPC, and 96.9% of those prescribed SPC received the subsidy (Table B of Section S8 in S1 Appendix).

No participant assigned to a usual care clinic sought treatment at an intervention clinic during the course of the study and vice versa.

### Primary outcome

At 24 months, the adjusted mean SBP (95% CI) in the intervention and usual care arms was 135.4 (133.1, 137.7) mmHg and 138.7 (136.4, 141.1) mmHg, respectively. The primary

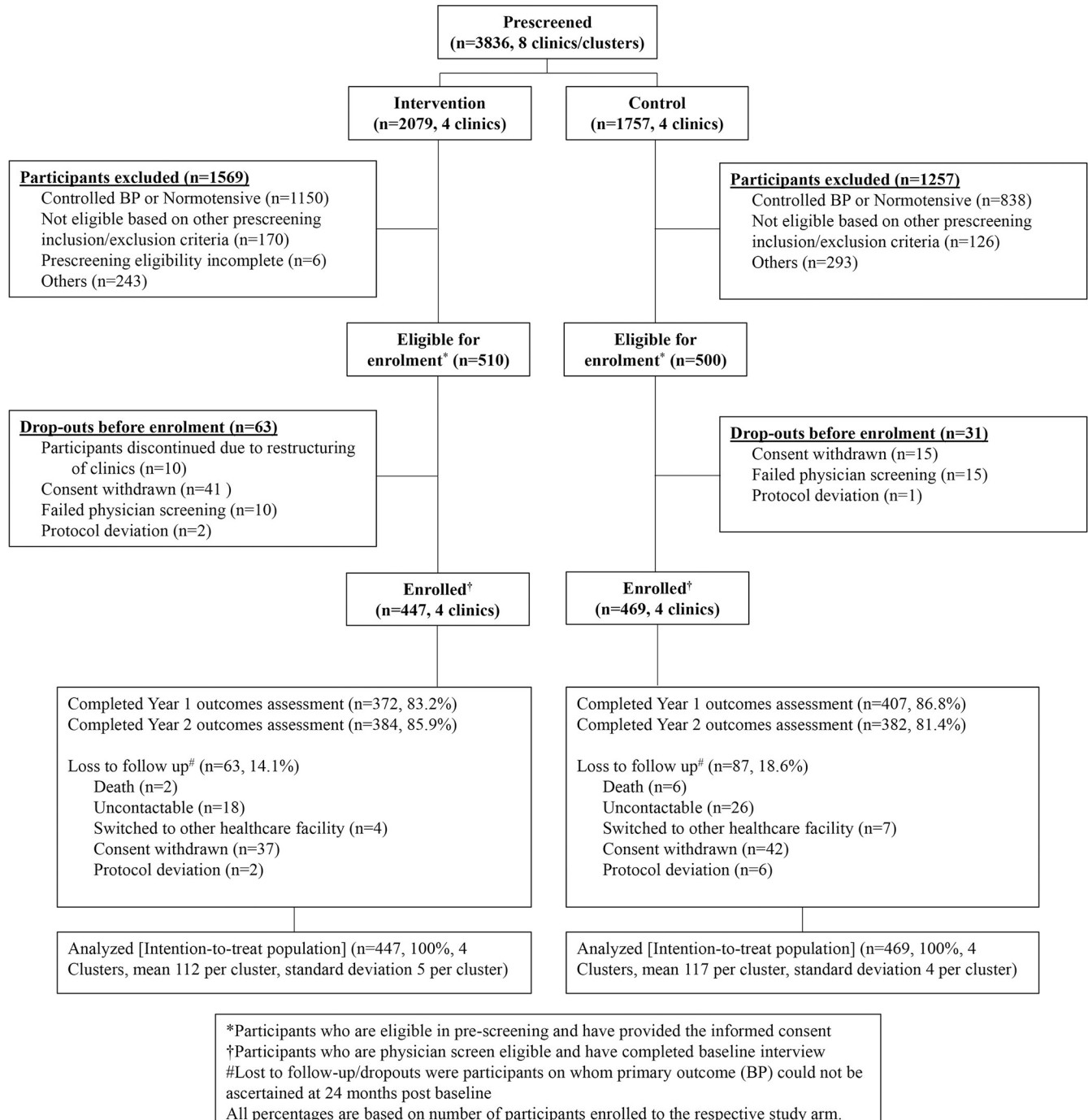

**Fig 1. Participant flow (CONSORT) chart.**

outcome, adjusted mean (95% CI) SBP, was significantly lower by −3.33 (−6.34, −0.32) mmHg (*p* = 0.03) in the intervention arm versus usual care (Fig 2A and Table 2).

At 24 months, the adjusted decline in SBP (95% CI) was −13.3 (−14.8, −11.8) in the intervention and −10.7 (−12.1, −9.2) in usual care. The adjusted decline (95% CI) in SBP from baseline was greater by −2.65 (−4.73, −0.57) mmHg (*p* = 0.01) in the intervention versus usual care (Table 3).

**Table 1. Baseline characteristics.**

| Characteristics[a] | Total (N = 916, 8 clinics) | All clinics (N = 916) | | P value |
| | | Multicomponent intervention (N = 447, 4 clinics) | Usual care (N = 469, 4 clinics) | |
|---|---|---|---|---|
| Age, mean (SD), years | 64.5 (9.8) | 63.0 (9.7) | 65.9 (9.7) | 0.41 |
| Female, n (%) | 454 (49.6) | 203 (45.4) | 251 (53.5) | 0.13 |
| Ethnicity, n (%) | | | | 0.88 |
| Chinese | 673 (73.5) | 334 (74.7) | 339 (72.3) | |
| Malay | 122 (13.3) | 57 (12.8) | 65 (13.9) | |
| Indian | 89 (9.7) | 36 (8.1) | 53 (11.3) | |
| Other | 32 (3.5) | 20 (4.5) | 12 (2.6) | |
| Secondary or higher education, n (%) | 670 (73.1) | 351 (78.5) | 319 (68.0) | 0.22 |
| Currently employed, n (%) | 445 (48.6) | 240 (53.7) | 205 (43.7) | 0.33 |
| Overweight or obese, n (%)[b] | 671 (73.3) | 324 (72.5) | 347 (74.0) | 0.80 |
| High waist circumference, n (%)[c] | 658 (71.8) | 308 (68.9) | 350 (74.6) | 0.39 |
| Chronic diseases | | | | |
| Self-reported heart disease, n (%) | 92 (10.0) | 33 (7.4) | 59 (12.6) | 0.24 |
| Self-reported stroke, n (%) | 37 (4.0) | 20 (4.5) | 17 (3.6) | 0.62 |
| Diabetes, n (%)[d] | 314 (34.3) | 138 (30.9) | 176 (37.5) | 0.45 |
| CKD, n (%)[e] | 351 (38.3) | 165 (36.9) | 186 (39.7) | 0.56 |
| Any of above chronic diseases, n (%)[f] | 548 (59.8) | 245 (54.8) | 303 (64.6) | 0.10 |
| BP, mmHg | | | | |
| Systolic, mean (SD) | 149.3 (13.0) | 148.4 (11.8) | 150.1 (14.0) | 0.30 |
| Diastolic, mean (SD) | 88.3 (9.8) | 89.4 (9.2) | 87.3 (10.2) | 0.26 |
| Current smoker, n (%) | 78 (8.5) | 40 (8.9) | 38 (8.1) | 0.88 |
| Physical activity, mean (SD), log MET-minute/week[g] | 6.1 (2.4) | 6.3 (2.4) | 6.0 (2.3) | 0.28 |
| 10-year calculated CVD FRS %, mean (SD) | 22.2 (7.6) | 21.7 (7.6) | 22.7 (7.5) | 0.34 |
| Ln urine albumin to creatinine (ACR), mean (SD) | 1.4 (1.0) | 1.3 (0.9) | 1.4 (1.0) | 0.31 |
| EQ-5D-5L VAS, mean (SD)[h] | 75.4 (13.4) | 77.0 (12.3) | 74.0 (14.2) | 0.50 |
| Currently on antihypertensive medications, n (%)[i] | | | | 0.29 |
| 0 | 26 (2.8) | 9 (2.0) | 17 (3.6) | |
| 1 | 408 (44.5) | 208 (46.5) | 200 (42.6) | |
| 2 | 315 (34.4) | 158 (35.3) | 157 (33.5) | |
| 3 or more | 154 (16.8) | 70 (15.7) | 84 (17.9) | |

[a]Baseline characteristics are represented as means (SD) or proportions (%) as appropriate.

[b]Defined as BMI $\geq$23.5 kg/m$^2$.

[c]High waist circumference was defined as waist circumference of $\geq$90 cm in males and $\geq$80 cm in females.

[d]Diabetes is defined as physician diagnosed or FBG $\geq$7 mmol/L or glycated hemoglobin (HbA1c) $\geq$6.5%.

[e]CKD is defined as urine ACR $\geq$3 mg/mmol or estimated glomerular filtration rate (eGFR) <60 ml/minute/1.73m$^2$.

[f]At least 1 comorbid chronic diseases of diabetes, CKD, self-reported heart disease, and self-reported stroke.

[g]Physical activity was assessed using the IPAQ score. The estimates are log MET-minute/week, higher score indicates higher activity level.The IPAQ Group, 2005 Guidelines.

[h]Health status was reported on a scale of 0 (worst imaginable health) to 100 (best imaginable health) using EQ-5D-5L VAS, where higher values indicate better health.

[i]A total of 13 participants had missing pharmacy data for antihypertensive medications.

ACR, albumin to creatinine ratio; BMI, body mass index; BP, blood pressure; CKD, chronic kidney disease; CVD, cardiovascular diseases; FBG, fasting blood glucose; FRS, Framingham risk score; IPAQ, International Physical Activity Questionnaire; Ln, natural logarithm; SD, standard deviation.

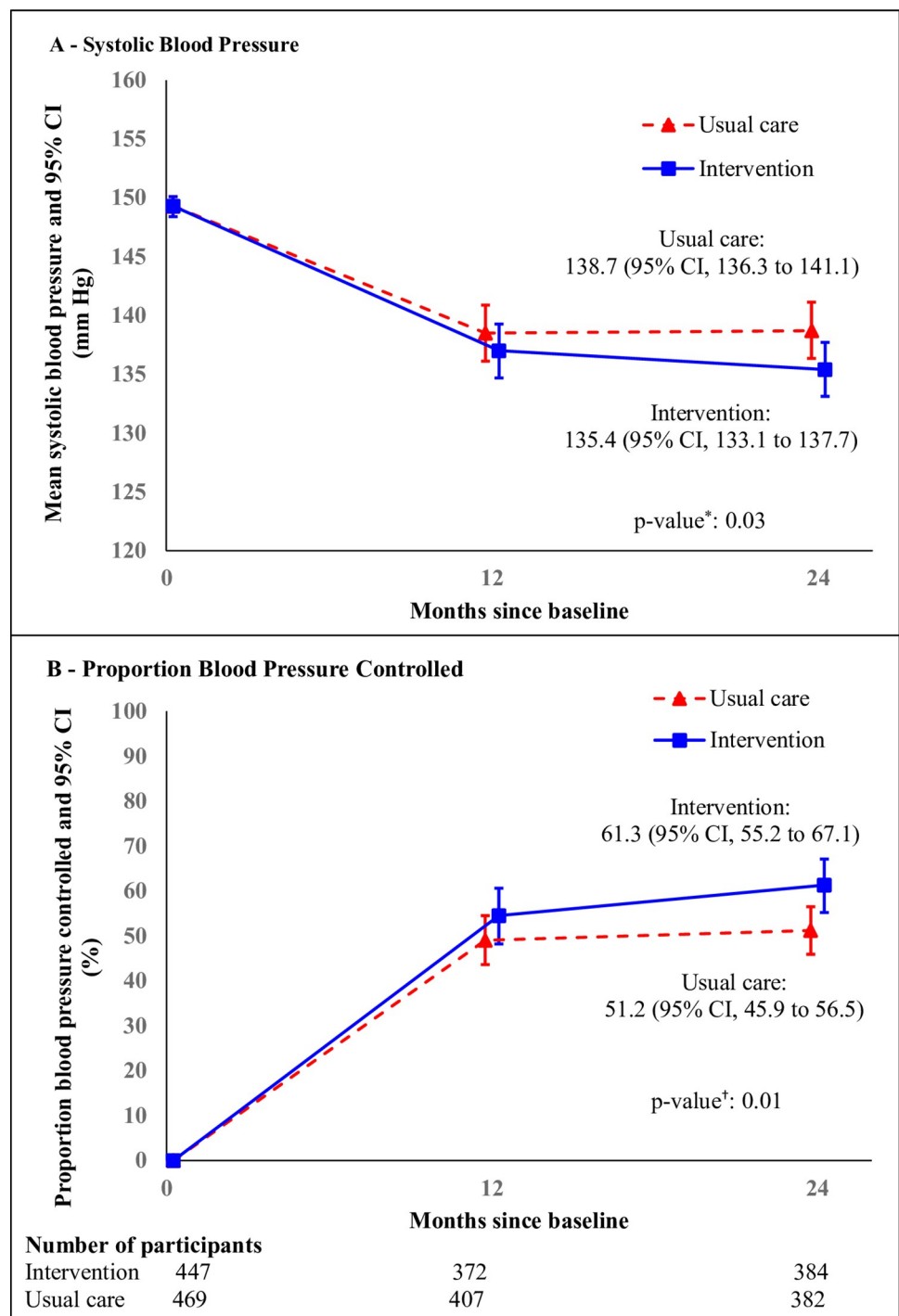

**Fig 2. Mean SBP and proportion BP controlled by treatment group over time.** Panel A: Mean SBP was estimated with a generalized linear MMRM for SBP, with cluster random effects for clinic, and random effects for participants. The I bars indicate 95% CIs. *The *p*-value for the difference in the mean SBP between 2 treatment groups at 24 months was 0.03. Panel B: Proportions BP controlled were estimated at 12 months and 24 months using a generalized linear mixed-effects model for BP control (SBP <140 mmHg and DBP <90 mmHg) as repeated measures, with fixed effect of baseline SBP, cluster random effects for clinic, and random effects for participants. The I bars indicate 95% CIs. †The *p*-value for the difference in the proportion BP controlled between 2 treatment groups at 24 months was 0.01. BP, blood pressure; CI, confidence interval; DBP, diastolic BP; MMRM, mixed model repeated measure; SBP, systolic BP.

**Table 2. Primary and BP-related key secondary outcomes by treatment group[a].**

| | n | Multicomponent intervention Mean (SE) (95% CI) (*n* = 447, 4 clinics) | n | Usual care Mean (SE) (95% CI) (*N* = 469, 4 clinics) | Adjusted mean difference[b]/OR (95% CI)[c] | *P* value |
|---|---|---|---|---|---|---|
| **Primary outcome** | | | | | | |
| SBP, mean (95% CI), mmHg | | | | | | |
| 24 months | 384 | 135.4 (0.98) (133.1, 137.7) | 382 | 138.7 (1.01) (136.4, 141.1) | −3.33 (−6.34, −0.32) | 0.03 |
| **Secondary outcomes** | | | | | | |
| DBP, mean (95% CI), mmHg | | | | | | |
| 24 months | 384 | 81.9 (0.86) (79.5, 84.3) | 382 | 81.4 (1.31) (77.5, 85.4) | 0.45 (−3.39, 4.29) | 0.78 |
| BP controlled to conventional goal (SBP <140 mmHg and DBP <90 mmHg), % (95% CI) | | | | | | |
| 24 months | 384 | 61.3 (11.0) (55.2, 67.1) | 382 | 51.2 (10.6) (45.9, 56.5) | 1.51 (1.10, 2.09)c | 0.01 |
| High FRS 10-year CVD risk, % (95% CI)[d] | | | | | | |
| 24 months | 363 | 47.3 (14.5) (40.3, 54.4) | 365 | 57.2 11.8 (51.4, 62.7) | 0.67 (0.47, 0.97) | 0.03 |
| Ln Urine ACR, mean (95% CI) | | | | | | |
| 24 months | 366 | 1.32 (0.07) (1.16, 1.48) | 355 | 1.53 (0.06) (1.39, 1.68) | −0.22 (−0.41, −0.02) | 0.03 |
| Number of antihypertensive medications per day, mean (95% CI) | | | | | | |
| 24 months | 393 | 1.9 (0.06) (1.8, 2.0) | 382 | 1.7 (0.07) (1.6, 1.9) | 0.18 (0.01, 0.36) | 0.04 |

[a]All analyses based on the ITT principle.

[b]Adjusted means differences between multicomponent intervention and usual care using repeated measures analysis.

[c]OR (95% CI).

[d]High 10-year FRS CVD risk score was defined as >20% risk of CVD at 10 years.

ACR, albumin to creatinine ratio; BP, blood pressure; CI, confidence interval; CVD, cardiovascular disease; DBP, diastolic BP; FRS, Framingham risk score; ITT, intention-to-treat; Ln, natural logarithm; OR, odds ratio; SBP, systolic BP.

## Key secondary outcomes

Most of the key secondary outcomes related to BP, vascular risk, and antihypertensive medication use showed improvements consistent with the primary outcome (Table 2). A greater percentage of participants achieved BP control at 24 months in the intervention (61.3%) arm versus usual care (51.2%) with associated odds ratio (95% CI) 1.51 (1.10, 2.09) (*p* = 0.01) (Fig 2B and Table 2). At 24 months, the percentage with high (>20%) 10-year CVD risk based on FRS was lower in the intervention arm with associated odds ratio (95% CI) 0.67 (0.47, 0.97) (*p* = 0.03) compared to usual care. Adjusted mean (95% CI) log albuminuria levels were lower by −0.22 (−0.41, −0.02) (log ACR) (*p* = 0.03) for intervention compared to usual care (Table 2). The number of antihypertensive medications used was higher by a mean (95% CI) of 0.18 (0.01, 0.36) (*p* = 0.04) per participant for intervention compared to usual care (Table 2) [25]. However, the mean DBP did not differ significantly between study arms (Table 2).

The findings for sensitivity analysis for SBP and BP-related key secondary outcomes were also consistent when analyzed using change from the baseline level (Table 3). Consistent results were obtained across all supplementary sensitivity analyses: per-protocol analysis, analysis restricted to participants completing the 24-month follow-up, analysis after accounting for clinically important baseline characteristics, multiple imputations analysis, and pre-COVID lockdown analysis (Table 3).

**Table 3. Sensitivity analyses of intervention effect on primary and key secondary outcomes.**

| | n | Multicomponent intervention Mean (SE) (95% CI) (N = 447, 4 clinics) | n | Usual care Mean (SE) (95% CI) (N = 469, 4 clinics) | Adjusted mean difference[a] (95% CI) | P value |
|---|---|---|---|---|---|---|
| **Change from Baseline in SBP and Key Secondary Outcomes[b]** | | | | | | |
| SBP, mmHg | | | | | | |
| 24 months | 384 | −13.3 (0.7) (−14.8, −11.8) | 382 | −10.7 (0.7) (−12.1, −9.2) | −2.65 (−4.73, −0.57) | 0.01 |
| DBP, mmHg | | | | | | |
| 24 months | 384 | −7.0 (0.4) (−7.9, −6.2) | 382 | −6.2 (0.7) (−7.1, −5.4) | −0.81 (−1.99, 0.37) | >0.05 |
| FRS 10-year CVD risk score[c] | | | | | | |
| 24 months | 359 | −2.3 (0.3) (−2.9, −1.8) | 357 | −1.3 (0.3) (−1.9, −0.8) | −1.00 (−1.77, −0.22) | 0.01 |
| Ln urine ACR | | | | | | |
| 24 months | 363 | 0.0 (0.0) (−0.1, 0) | 353 | 0.1 (0.1) (0.0, 0.2) | −0.17 (−0.31, −0.02) | 0.02 |
| Number of antihypertensive medications per day | | | | | | |
| 24 months | 393 | 0.2 (0.1) (0.1, 0.4) | 382 | 0 (0) (0, 0.1) | 0.20 (0.07, 0.34) | 0.01 |
| **SBP using MMRM[d]** | | | | | | |
| Per protocol analysis[e] | | | | | | |
| 24 months | 325 | 134.8 (1.0) (132.6, 137.0) | 382 | 138.7 (1.0) (136.3, 141.2) | −3.94 (−6.93, −0.94) | 0.01 |
| Restricted to patients completing 24-month follow-up[f] | | | | | | |
| 24 months | 384 | 135.3 (0.9) (133.1, 137.5) | 382 | 138.7 (0.8) (136.9, 140.6) | −3.43 (−6.07, −0.80) | 0.01 |
| Final analysis adjusting for clinically important variables at baseline[g] | | | | | | |
| 24 months | 384 | 135.7 (0.8) (133.8, 137.6) | 382 | 138.2 (1.0) (136, 140.4) | −2.53 (−5.00, −0.09) | 0.04 |
| After multiple imputation[h] | | | | | | |
| 24 months | 447 | 136.3 (1.0) (134.2, 138.4) | 469 | 139.8 (1.0) (137.9, 141.7) | −3.39 (−6.32, −0.65) | 0.02 |
| After restricting to participants followed before COVID-19[i] | | | | | | |
| 24 months | 322 | 134.9 (1.1) (132.3, 137.4) | 372 | 138.9 (1.0) (136.6, 141.2) | −4.00 (−7.14, −0.86) | 0.02 |

[a]Adjusted means differences between multicomponent intervention and usual care using repeated measures analysis.

[b]Change from baseline in each outcome (at year-1 and year-2) was analyzed at the participant level after adjusting for baseline values of respective outcome as a fixed covariate. The analyses were performed based on the ITT principle.

[c]Ten-year FRS CVD risk score.

[d]Primary outcome of SBP at baseline, 12 and 24 months was modeled at the patient level in a likelihood-based, linear MMRM analysis with cluster random effects for clinic.

[e]Participants who are high CVD risk as per the checklist and did not receive SPC in the multicomponent intervention (n = 73) as per protocol were excluded from this analysis.

[f]Participants who did not complete the year-2 outcomes assessments (n = 150) were excluded from this analysis.

[g]Primary outcome analysis at the participant level after adjusting for important baseline characteristics as covariates. The characteristics included as covariates in adjustments were age, gender, waist circumference, diabetes, and 10-year FRS CVD risk score.

[h]Primary outcome analysis after multiple imputations to replace missing values of SBP at 1 year: 137 (15.0%) and 2 years: 150 (16.4%).

[i]SBP readings of participants followed at year-2 outcomes assessments on or after 12 March 2020 (n = 72, intervention– 62, and usual care– 10) were excluded from the analysis to assess the effect of multicomponent intervention before COVID-19 was announced as pandemic by WHO.

ACR, albumin to creatinine ratio; BP, blood pressure; CI, confidence interval; COVID-19, Coronavirus Disease 2019; CVD, cardiovascular disease; DBP, diastolic BP; FRS, Framingham risk score; ITT, intention-to-treat; Ln, natural logarithm; MMRM, mixed-effects model repeated measures; OR, odds ratio; SE, standard error; SBP, systolic BP; SPC, single pill combination; WHO, World Health Organization.

### Other secondary outcomes

Secondary outcomes related to health behaviors and quality of life also showed incremental improvements in response to the intervention (Table C of Section S8 in S1 Appendix). Self-reported overall health status was better in the intervention arm. The mean score (95% CI) on the EQ-5D-5L VAS in the intervention arm was higher than usual care by 6.28 (0.56, 11.99) points ($p = 0.03$). Secondary outcomes of BMI, waist circumference, current smokers, cholesterol, and estimated GFR were not significantly different between study arms ($p > 0.20$ for all) (Supporting information Table C).

The proportion of participants with incident SAEs did not differ significantly between study arms. Total mortality was not different between intervention (0.5%) compared to usual care (1.3%) ($p = 0.29$). Fewer participants reported injuries or falls with intervention compared to usual care (0.5 versus 2.8%, respectively) ($p = 0.01$) (Table D of Section S8 in S1 Appendix).

Effects of intervention on the primary outcome of SBP were consistent for all predefined subgroups, with no evidence of effect heterogeneity (Fig 3).

The intraclass cluster coefficient was 0.009 for SBP and 0.02 for change in SBP, respectively (Table E of Section S8 in S1 Appendix). The raw means of primary and key secondary outcomes were consistent with the adjusted means. (Table F of Section S8 in S1 Appendix).

### Cost of intervention

The incremental cost of intervention delivery per eligible participant was SGD $231 (USD $170 annually, exchange rate 1 SGD $ = 0.74 USD $ as of January 2022) (Table G of Section S8 in S1 Appendix).

## Discussion

In this cluster-randomized trial in adults with uncontrolled hypertension visiting primary care clinics in Singapore, a multicomponent intervention incorporating risk-based treatment by trained physicians, subsidized SPC medications, nurse-led motivational conversation, and telephone follow-ups, significantly lowered SBP and improved BP control at 24 months, compared to usual care. The intervention also increased use of antihypertensive medications, lowered cardiovascular risk and albuminuria, improved some aspects of health-related quality of life, and there were no safety concerns. The potential ease of adaptation into the existing primary care system and low cost (USD $14 per participant per month) indicate rapid scalability and sustainability of the intervention in Singapore and other countries with similar infrastructure.

The vast majority of previous multicomponent intervention trials demonstrating BP lowering effectiveness in high-income countries were of short duration (median, 6 months) [11]. A strategy of ACEI (Lisinopril)-hydrochlorothiazide SPC medication-based therapy was shown effective in California adults [34]. Trials of interventions in the United States that included telephone follow-ups did not show persistent intervention benefits beyond 12 months [12,13]. However, those trials did not include risk-based management with SPC medications or any related subsidies. Trials longer than 1 year were mostly from low- and middle-income countries (LMICs) with primary care infrastructures differing from Singapore and other high-income countries [35,36].

Our intervention strategy departs from previous trials in its composition as a combination of risk-based management and subsidized SPC medications coupled with proactive motivational counseling delivered via the routine primary care infrastructure.

An additional feature of our trial was a prespecified assessment of cardiovascular risk and albuminuria outcomes—both robust prognostic indicators of CVD [37]. Our trial was not

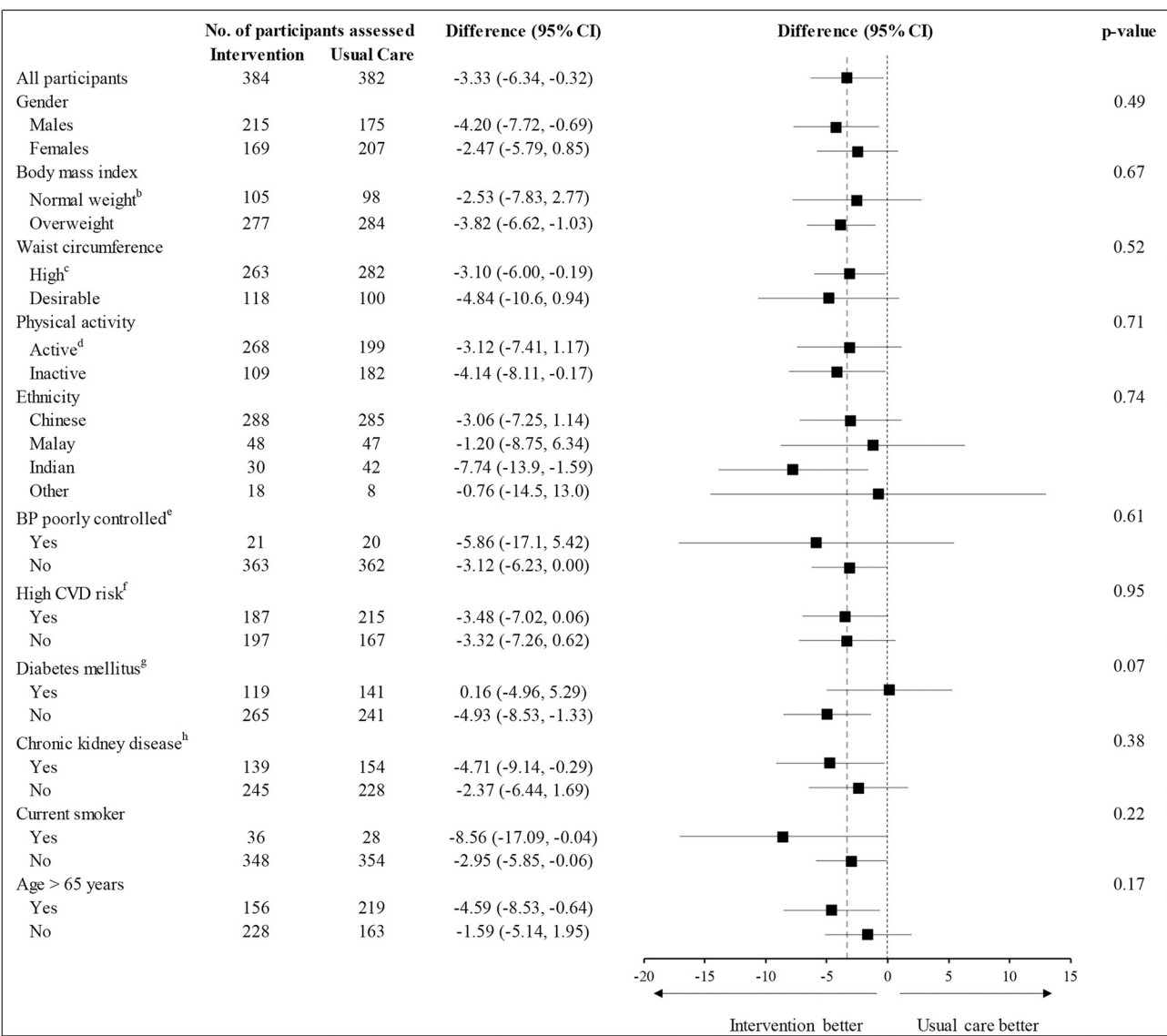

| | No. of participants assessed | | Difference (95% CI) | Difference (95% CI) | p-value |
|---|---|---|---|---|---|
| | Intervention | Usual Care | | | |
| All participants | 384 | 382 | -3.33 (-6.34, -0.32) | | |
| Gender | | | | | 0.49 |
| Males | 215 | 175 | -4.20 (-7.72, -0.69) | | |
| Females | 169 | 207 | -2.47 (-5.79, 0.85) | | |
| Body mass index | | | | | 0.67 |
| Normal weight[b] | 105 | 98 | -2.53 (-7.83, 2.77) | | |
| Overweight | 277 | 284 | -3.82 (-6.62, -1.03) | | |
| Waist circumference | | | | | 0.52 |
| High[c] | 263 | 282 | -3.10 (-6.00, -0.19) | | |
| Desirable | 118 | 100 | -4.84 (-10.6, 0.94) | | |
| Physical activity | | | | | 0.71 |
| Active[d] | 268 | 199 | -3.12 (-7.41, 1.17) | | |
| Inactive | 109 | 182 | -4.14 (-8.11, -0.17) | | |
| Ethnicity | | | | | 0.74 |
| Chinese | 288 | 285 | -3.06 (-7.25, 1.14) | | |
| Malay | 48 | 47 | -1.20 (-8.75, 6.34) | | |
| Indian | 30 | 42 | -7.74 (-13.9, -1.59) | | |
| Other | 18 | 8 | -0.76 (-14.5, 13.0) | | |
| BP poorly controlled[e] | | | | | 0.61 |
| Yes | 21 | 20 | -5.86 (-17.1, 5.42) | | |
| No | 363 | 362 | -3.12 (-6.23, 0.00) | | |
| High CVD risk[f] | | | | | 0.95 |
| Yes | 187 | 215 | -3.48 (-7.02, 0.06) | | |
| No | 197 | 167 | -3.32 (-7.26, 0.62) | | |
| Diabetes mellitus[g] | | | | | 0.07 |
| Yes | 119 | 141 | 0.16 (-4.96, 5.29) | | |
| No | 265 | 241 | -4.93 (-8.53, -1.33) | | |
| Chronic kidney disease[h] | | | | | 0.38 |
| Yes | 139 | 154 | -4.71 (-9.14, -0.29) | | |
| No | 245 | 228 | -2.37 (-6.44, 1.69) | | |
| Current smoker | | | | | 0.22 |
| Yes | 36 | 28 | -8.56 (-17.09, -0.04) | | |
| No | 348 | 354 | -2.95 (-5.85, -0.06) | | |
| Age > 65 years | | | | | 0.17 |
| Yes | 156 | 219 | -4.59 (-8.53, -0.64) | | |
| No | 228 | 163 | -1.59 (-5.14, 1.95) | | |

**Fig 3. Subgroup analyses for SBP at 24 months, according to participant characteristics at baseline[a].** [a]Subgroups defined based on the status of the participants at the time of baseline interview. [b]Normal weight is defined as BMI <23.5 kg/m² per Asian cutoffs. [c]High waist circumference was defined as waist circumference of ≥90 cm in males and ≥80 cm in females. [d]Physical activity was assessed using the IPAQ score. The IPAQ Group, 2005. [e]Defined as SBP ≥160 mmHg and DBP ≥100 mmHg. [f]Defined using the standardized physician management checklist developed for the study. The participant is categorized as high CVD risk if any of the following criteria is met: high CVD risk by FRS adapted for Singapore ≥20, diabetes, ACR >34 mg/mmol, eGFR <60 ml/minute/1.73 m², physician diagnosed heart disease, and self-reported stroke. [g]Diabetes is defined as physician diagnosed or FBS >7 mmol/L or glycated hemoglobin (HbA1c) >6.5%. [h]CKD is defined as eGFR <60 ml/minute/1.73 m² or urine ACR ≥3 mg/mmol. ACR, albumin to creatinine ratio; BMI, body mass index; BP, blood pressure; CVD, cardiovascular disease; DBP, diastolic BP; eGFR, estimated glomerular filtration rate; FBS, fasting blood sugar; FRS, Framingham risk score; IPAQ, International Physical Activity Questionnaire; SBP, systolic BP.

designed to unravel how much of the benefit of the multicomponent intervention was attributable to each individual component. It is likely that the intervention benefit on BP observed in our trial is a reflection of positive interaction among its components, which addressed the barriers to hypertension care at the patient, physician, and health systems levels. First, trained intervention-arm physicians are more likely to prescribe antihypertensive medications, intensify antihypertensive treatments, and prescribe SPC medications [38]. SPC medications are better tolerated than higher doses of individual medications, which improves adherence [8,9].

Second, the motivational conversation and telephone follow-ups likely enhanced uptake of healthy lifestyles (potentially physical activity) and medications. Third, the subsidy component of the intervention likely served as an additional stimulus to SPC medication utilization by reducing the copayment by the patients at high risk of CVD [39].

We found that although BP declined in both groups at 12 and 24 months, the benefit of intervention on lowering BP was apparent only at the latter time point. The longevity of the benefit of intervention at 24 months is unique for high-income countries. These findings are consistent with our previous work in LMICs and possibly reflect synergies among the intervention components over time [35]. For example, messages on adherence to lifestyle and anti-hypertensive medications were reinforced during repeated telephone-based follow-ups by trained nurses. In addition, the patients identified with elevated BP were flagged for the clinic visit when trained physicians escalated antihypertensive drugs per the study algorithm, including SPC antihypertensive medications for high-risk patients who were incentivized by the subsidy [29,40]. Nurse and physician competencies may have enhanced over time with the annual retraining.

Hypertension is an immense public health challenge, implicated in over 50% of heart disease, stroke, and heart failure cases [41]. BP control remain suboptimal globally even in high income countries [1]. Our findings regarding benefits of a multicomponent intervention leveraging the existing healthcare infrastructure have implications for clinical practice and policy globally. Discussions with the health departments and advisory committees are ongoing for scaling-up the intervention across primary care clinics in Singapore.

Our intervention is a potentially viable option for many Asian countries such as Japan, Korea, Taiwan, and urban China with equally high-risk populations and hypertension-related CVD burden, similar public health infrastructure with wide availability of generic SPC antihypertensive medications [42]. Looking beyond Asia, the potential impact of the intervention could be even more significant in the US, Canada, UK, and many European countries where recent trends indicate a plateauing or rise in uncontrolled hypertension rates [43,2,44–46]. Our intervention addressed barriers to hypertension care at multiple levels, yet it was relatively straightforward. In addition, the implementation fidelity indicated excellent uptake of strategies (patients, physicians, nurses, and clinic) levels demonstrating successful leveraging of the existing healthcare infrastructure. Therefore, it would be logistically possible to scale up the intervention rapidly with marginal costs. Moreover, the physician's and nurses' staffing ratios relative to the population (2.4 and 6.2 per 1,000 population, respectively) and the electronic health record system in Singapore's public sector are comparable to healthcare systems in high- and upper-middle-income countries [47]. Generic formulations of antihypertensive medications, including SPC drugs, are available [10]. Various forms of vouchers, which have been used to incentivize care in other diseases [48,49], could substitute for our use of a subsidy for the SPC.

We suggest that suitably modified forms of a multicomponent intervention such as ours be evaluated in other settings and their effects on direct measures of vascular disease. The major strengths of our trial are the cluster-randomized design, intervention delivered with high fidelity using the existing public health infrastructure that minimized cost and enhances sustainability and generalizability, and 2-year follow-up. Consistent results of ITT, per-protocol, and sensitivity analyses at both the clinic and patient level for BP, CVD risk, and albuminuria indicate the robustness of our findings.

Our trial has limitations. First, since this was a pragmatic trial, it was not possible to mask the clinical practice teams to the random group assignment. Moreover, the patients were aware of the group to which their practice was assigned at enrollment. However, the key

outcome (BP) was measured by using an automated device, which minimized the risk for biased outcome assessment. Likewise, the assessment of several vascular risk factors including fasting glucose, lipids, and albuminuria were objectives measures obtained directly from the laboratory and medication data from pharmacy records. Thus, the potential of reporting bias for the primary and key secondary outcomes is minimal. Second, almost 16% of the trial cohort was lost to follow-up during 2 years. However, retention was considered very good, given the COVID-19 pandemic restrictions imposed during the last 3 months of follow-up. Moreover, sensitivity analysis restricted to participants before COVID-19-related lockdown, as well as multiple imputation accounting for missing values yielded consistent results. Third, our trial did not have sufficient power to detect the intervention effect on several secondary outcomes especially DBP and lifestyle measures. However, several large-scale epidemiological studies and systematic reviews of trials have shown that SBP—our primary outcome—is a much stronger predictor of CVD morbidity and mortality than DBP [50,51]. Moreover, the intervention improved BP control, accounting for both SBP and DBP target levels. It is also possible that physicians were not aggressive with titrating therapy to the DBP target in some patients, perhaps due to concerns with excessive lowering of DBP levels and paradoxical increase in CVD [52]. A post-intervention acceptability study is planned to explore these factors. Fourth, the trial duration was insufficient to assess cardiovascular events. However, even a 2-mmHg decline in SBP has been associated with a 7% to 10% decrease in deaths from coronary disease and stroke [51,53–55]. Moreover, improvement in the secondary outcomes of CV risk and albuminuria—albeit multiplicity unadjusted—indicate potential in the downstream benefit of the strategy on vascular risk reduction. Fifth, the benefit of the intervention on albuminuria reduction could be a chance association, pending further studies for confirmation. However, blockers of the renin-angiotensin-aldosterone system (RAAS) such as losartan are known to have antiproteinuric effects that are in addition to their BP-lowering effect, especially when used together with diuretics [56,57]. RAAS blockers and diuretics were part of the SPC medications used more frequently in patients in the intervention group. Sixth, the existence of some elements of the intervention among control clinics, or contamination, remains a possibility with implications of a diluted intervention effect. Moreover, usual care participants also experienced a decline in BP, which was most likely due to behavior modification in response to BP screening by the CRCs. These factors would contribute to underestimation of the full benefit of the intervention. Moreover, regression to the mean could account for some reduction in BP, although it would apply equally to both arms [56,58]. Finally, the number of clusters was relatively small, which increases the possibility of randomization imbalance at baseline. However, we noted no statistically or clinically meaningful differences in any baseline characteristics between the randomized groups. Furthermore, we employed a contemporary analytic approach with no assumptions for baseline [30,31]. Moreover, sensitivity analysis adjusting for baseline characteristics for the primary outcome yielded consistent results. Thus, we believe our results are robust.

## Conclusions

In conclusion, a multicomponent intervention, including physician training on risk-based subsidized SPC antihypertensive medications, nurse-led motivational conversation, and proactive telephone follow-ups, led to reduced SBP and improved BP control over 24 months among patients with uncontrolled hypertension in Singapore's primary care clinics. Given the potential ease of adaptation and rapid scalability into many existing health systems at marginal costs, implementation of our strategy would represent a step forward in the high-priority global effort to reduce hypertension-related morbidity and death.

## Supporting information

**S1 CONSORT Checklist. CONSORT 2010 checklist of information to include when reporting a cluster randomized trial.**
(DOCX)

**S1 Appendix. Supplementary Appendix: S1 Protocol S1 Statistical Analysis Plan.** Section S1: Research Contributions. Section S2: Nurse telephone follow-up checklist. Section S3: Physician Management Checklist. Section S4: Summary of intervention training. Section S5: Anti-hypertensive treatment algorithm. Section S6: Statistical methods for sensitivity for systolic blood pressure (BP). Section S7: Program delivery cost estimation—methods, assumptions, and data sources. Section S8: Tables A to G. Section S9. References.
(DOCX)

## Acknowledgments

We acknowledge the contribution of all investigators, coordinators, staff, and leadership of the SingHypertension Study at the respective institutions including at SingHealth Polyclinics, National University Polyclinics, and Duke-NUS Medical School, Singapore. A list of investigators, coordinators, and staff from all participating clinics is provided in the Supporting information.

Finally, we thank all the SingHypertension trial participants, as the trial would not have been possible without their cooperation.

An independent Data Safety Monitoring Board (DSMB) met 6 times to review the quality and safety.

The trial abstract was presented in the Late-breaking "Latest Science in Hypertension" session at the European Society of Cardiology Congress August 27 to 29, 2021.

THJ had full access to all the data in the study and takes responsibility for the integrity of the data and the accuracy of the data analysis.

### Members of DSMB

Dr. Tan Ru San (chair), National Heart Centre, Singapore; Professor Doris Young, Melbourne Medical School, Australia; Professor Vathsala Anantharaman, National University Hospital, Singapore; and Dr. Edwin Chan Shih Yen, Singapore Clinical Research Institute, Singapore.

### Participating clinics

Bukit Merah, Bedok, Marine Parade, Outram, Pasir Ris, Queenstown, Seng Kang, Tampines polyclinics.

## Author Contributions

**Conceptualization:** Tazeen Hasan Jafar.

**Data curation:** Tazeen Hasan Jafar, Rupesh Madhukar Shirore, Chris Wan Teng Goh, Reena Chandhini Subramanian, Chandrika Ramakrishnan, Jianying Liu.

**Formal analysis:** Rupesh Madhukar Shirore, John Carson Allen.

**Funding acquisition:** Tazeen Hasan Jafar.

**Investigation:** Tazeen Hasan Jafar, Ngiap Chuan Tan, John Carson Allen, Eric Andrew Finkelstein, Siew Wai Hwang, Agnes Ying Leng Koong, Peter Kirm Seng Moey, Gary Chun-Yun Kang, Anandan Gerard Thiagarajah.

**Methodology:** Tazeen Hasan Jafar, John Carson Allen.

**Project administration:** Tazeen Hasan Jafar, Ngiap Chuan Tan, Rupesh Madhukar Shirore, Chris Wan Teng Goh, Reena Chandhini Subramanian, Anandan Gerard Thiagarajah, Chandrika Ramakrishnan, Jianying Liu.

**Supervision:** Tazeen Hasan Jafar, Ngiap Chuan Tan, Rupesh Madhukar Shirore, Siew Wai Hwang, Agnes Ying Leng Koong, Peter Kirm Seng Moey, Gary Chun-Yun Kang, Anandan Gerard Thiagarajah, Chandrika Ramakrishnan.

**Validation:** Ching Wee Lim.

**Visualization:** Rupesh Madhukar Shirore.

**Writing – original draft:** Tazeen Hasan Jafar.

**Writing – review & editing:** Tazeen Hasan Jafar, Ngiap Chuan Tan, Rupesh Madhukar Shirore, John Carson Allen, Eric Andrew Finkelstein, Siew Wai Hwang, Agnes Ying Leng Koong, Peter Kirm Seng Moey, Gary Chun-Yun Kang, Chris Wan Teng Goh, Anandan Gerard Thiagarajah, Chandrika Ramakrishnan, Jianying Liu.

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
