## [Editor Report · Decision Letter 0]

16 Feb 2022

Dear Dr Jafar, 

Thank you for submitting your manuscript entitled "Integration of a Multicomponent Intervention for Hypertension into Primary Healthcare Services - A Cluster Randomised Controlled Trial" for consideration by PLOS Medicine.

Your manuscript has now been evaluated by the PLOS Medicine editorial staff and I am writing to let you know that we would like to send your submission out for external peer review.

Please re-submit your manuscript within two working days, i.e. by Feb 18 2022 11:59PM.

Kind regards,

Callam Davidson

Associate Editor

PLOS Medicine

---

## [Decision Letter · Decision Letter 1]

4 Apr 2022

Dear Dr. Jafar,

Thank you very much for submitting your manuscript "Integration of a Multicomponent Intervention for Hypertension into Primary Healthcare Services - A Cluster Randomised Controlled Trial" (PMEDICINE-D-22-00512R1) for consideration at PLOS Medicine. 

Your paper was evaluated by an associate editor and discussed among all the editors here. It was also discussed with an academic editor with relevant expertise, and sent to independent reviewers, including a statistical reviewer. The reviews are appended at the bottom of this email and any accompanying reviewer attachments can be seen via the link below:

[LINK]

In light of these reviews, I am afraid that we will not be able to accept the manuscript for publication in the journal in its current form, but we would like to consider a revised version that addresses the reviewers' and editors' comments. Obviously we cannot make any decision about publication until we have seen the revised manuscript and your response, and we plan to seek re-review by one or more of the reviewers. 

We hope to receive your revised manuscript by Apr 25 2022 11:59PM. Please email us (plosmedicine@plos.org) if you have any questions or concerns.

We look forward to receiving your revised manuscript. 

Sincerely,

Callam Davidson, 

Associate Editor

PLOS Medicine

plosmedicine.org

Comments from the Academic Editor:

This is a well conducted, pragmatic study of a multifaceted intervention. I think the effect size is in keeping with pragmatic health care interventions of these nature and so I am not that concerned by reviewer 4 comments on that particular issue. The longer follow-up is certainly a value add as the authors point out most studies are of much shorter duration.

I do agree with reviewer 2 that the major gap in the paper is more information on implementation fidelity. There is absolutely no information on numbers of people reached by each of the intervention components including medication use and adherence rates and whether fidelity changed over time which could help explain why effects emerged in year 2 (assuming much of the year 1 reduction was regression to the mean). It really is a bit of a black box trying to work out the causal path to effectiveness and this does not help readers with generating new knowledge or replicating the study in other settings (as recommended by the authors).

The cost analysis is cursory. It only assesses costs of the intervention and does not include costs of care utilisation, medications, diagnostic tests etc. The EQ5D-5L VA scores only are reported which shows a difference but the mean scores are not reported. It is difficult to see how this intervention might generate a quality of life gain in such a short period of time with just a small reduction in BP. That said a cost per QALY estimate could be relatively easily generated from the data available to determine an incremental cost effectiveness ratio and I suggest this be included in the paper to help us understand this in more detail.

A few other issue of note:

1. I also was quite perplexed by the large rise in albuminuria in the usual care group. This is quite a spurious finding and not at all in keeping with the effect sizes observed in the other outcome variables. This warrants greater attention.

2. I was also surprised that other cardimetabolic parameters such as lipids and blood glucose were not included. This information was collected and lipids are stated as a secondary outcome measure on p.11 but it is not reported. I have not checked the protocol and SAP but worth checking if all outcomes measures are reported as per protocol.

3. The authors state total mortality was lower for intervention (0.5%) compared to usual 437 care (1.3%) (p=0.29) - this should be amended as there was no statistically significant difference in mortality

Please structure your abstract using the PLOS Medicine headings (Background, Methods and Findings, Conclusions).

Please report your abstract according to CONSORT for abstracts, following the PLOS Medicine abstract structure (Background, Methods and Findings, Conclusions) http://www.consort-statement.org/extensions?ContentWidgetId=562

Abstract Background: Provide the context of why the study is important. The final sentence should clearly state the study question.

Abstract Methods and Findings:

Please state that there was no masking in the trial.

Please define both the intervention and control states.

Please provide the number of participants lost to follow up in each group.

Please indicates the dates during which study enrollment and follow up occurred.

Please include the important dependent variables that are adjusted for in the analyses.

Please include a summary of adverse events.

Please ensure relevant p values are reported.

In the last sentence of the Abstract Methods and Findings section, please describe the main limitation(s) of the study's methodology.

The Data Availability Statement (DAS) requires revision. If the data are not freely available, please describe briefly the ethical, legal, or contractual restriction that prevents you from sharing it. Please also include an appropriate contact (web or email address) for inquiries (this cannot be a study author).

Thank you for including an Author Summary. Please update as follows:

* Relocate such that Author Summary is placed after Abstract and before Introduction.

* Combine bullets one and two in a single-sentence bullet.

* In the final bullet under ‘Why was this study done?’, please replace ‘were’ with ‘are’ and relocate the comma after ‘mixed’ to come after ‘(majority <6 months)’. 

* Bullets under the ‘What did the researchers do and find?’ can be slightly more succinct (aim for 3 single sentence bullet points, if possible). 

* Please quantify the main findings (i.e. lowering of blood pressure and improvement in BP control relative to usual care) and consider stating ‘was not associated with any safety concerns’ rather than ‘was safe’ (same comment applies at Line 471). 

Please remove funding details from the first page (this information is captured in your Financial Disclosure statement in the submission form).

Citations should be in square brackets and preceding punctuation.

Thanks for completing the CONSORT checklist. Please cite your checklist in the Methods (‘This trial is reported according to the Consolidated Standards of Reporting Trials (S1 Checklist)’ at line 224. 

Please update the checklist to use section and paragraph numbers, rather than page numbers.

Please define what is meant by dropouts/loss to follow up.

Please present the safety data for the study including numbers of specific events and whether or not adverse events are thought to be related to treatment.

Please indicates the dates during which study enrollment and follow up occurred.

Line 217: Please cite your protocol as an item within the Supporting Information (see our guidance here https://journals.plos.org/plosmedicine/s/supporting-information). 

Line 210: Outcomes could be described more clearly (they are easier to determine in the protocol/on clinicaltrials.gov).

 Line 271: Please cite the SAP you have included in your Supporting Information.

Line 281: Should Sensitivity Analyses have their own subsection?

Given that randomisation has been performed, please remove p values from Table 1 and Table S1.

Please clarify the difference between Table 1 and Table S1.

Please ensure consistency when reporting p values in terms of decimal places (e.g. Table 3).

Line 433: Should this not be P>0.05 for all given threshold for significance?

Line 437: Why is this reported as a difference when P=0.29?

Supplementary sections S1-4 are not cited in the text as far as I can see?

Please remove funding details from the Acknowledgements. 

The Data Availability Statement can be removed from the end of the main text – please ensure all relevant information is captured in the relevant question on the Submission Form.

Comments from the reviewers:

Reviewer #1: Prof. Tazeen H Jafar and his colleague conducted the trail to assess the effectiveness of a multicomponent intervention on blood pressure control compared to usual care, and the findings showed a significant lowering of SBP level and a higher control rate of BP in management group than that in usual group. Several comments:

1. How about the comparability of the Clinics between two groups? What`s the proportion of education degree of the physicians, how many years did they work, how about the social economic level, etc.

2. Although four trainings (beginning, 3m, 12m, 24m) for doctors, three times (beginning, 12m, 24m) for nurses, were done in intervention group, except the last one, only 3 times for doctors, 2 times for nurses, if there were the other effect beside the "intervention". Subsidies on drug?

3. According to the design, there was no power to test the effect on the secondary outcomes, why added many items?

Reviewer #2: This was a cluster-based randomized program which compared multicomponent intervention vs usual care for control of BP in Singapore setting. In a nutshell, 8 clinics participated, 4 were intervention clinics and 4 were usual care clinics. The intervention included training physicians to stratify risk, 50% subsidy for fixed dose combination (FDC) of ARB and thiazide, motivational interview by nurses and regular telephone calls. A clinical research coordinator also performed evaluation at month 0, 12 and 24 in both groups. The intervention group was supposed to be seen by doctors/nurses at 3-4 monthly intervals. Over 400 patients were recruited in each group and the endpoint was SBP reduction at 24 months along with Framingham 10-year risk score and ACR. There was also mention of EQ5D evaluation and cost effectiveness analysis. About 80% of patients returned at month 24 for evaluation. The study was influenced by the COVID-19 outbreak. 

The analysis was very technical using multiple statistical methods to compare clinic-based rather than individual patient-level data with adjustment for clusters and time trends. Analyses were also performed using imputed data with multiple modelling as well as using data of patients who returned at month 24 in a per protocol analysis. The conclusion was the intervention group achieved lower SBP by 3 mmHg and had an 1.5 odds ratio of achieving a BP<140/90 with 40% risk reduction of 10-year CVD risk score and reduction in ACR compared with usual care. 

There is now consensus that the challenges in control of NCD lie in implementation which needs support from healthcare planners and payors in addition to providers and patients. Multicomponent interventions are also well accepted although its implementation depends on context which needs to take into consideration the usual care model for scalability and sustainability. It is here that the paper lacks granularity as to what the usual care model is. Do people in Singapore have a regular family doctor or do they shop around for doctors? For WHO defined essential lab tests (e.g. lipid, A1c, RFT) and medications (statins, ACEi, metformin, SU etc), do they have to co-pay or are they fully covered? Is there a ceiling for government coverage and do patients have to pay every time out of pocket to see their doctor or a nurse? Do patients get medications from the clinic or pharmacy and can they self purchase medications without a prescription? Are there huge differences amongst these clinics in terms of staff to patient ratio, care pathways and payment structure? All these factors may influence the quality of care in the intervention and usual care groups.

In its current format, it is challenging to visualize the implementation procedures and patient/provider adherence as well as pre and end of study clinical profiles in both groups. Apart from BP and some demographic data, there was no information on eGFR, lipids and glycaemic indexes, some of which were needed to compute the 10-year CVD risk score. There was mention of EQ5D but not reported in the paper. The cost-effectiveness analysis was very minimal and it is not clear what was the average number of man-hour involved (doctors/nurses), number of face to face visits and telephone consultations per patient during this 2-year program in the intervention group. What was the no of clinic visits in the usual care group? These information should have been captured by the project coordinator, indeed, what information was collected during these important evaluation visits at 0, 12 and 24 month. 

The authors proposed that the model is applicable to many countries. If this is their message, then they have to be very specific on the implementation details. How many nurses and doctors were involved in each of these participating clinics? What is the provider:patient ratio? What were the drop out and adherence rates and why did people drop out? Why did patients do not join the program?

From a clinical perspective, were there any changes in other medications for lipids, BP and glucose? What were the changes in these risk factors that contribute to the 40% reduction in 10-year CVD risk score? Was there any difference in new cases of diabetes between the 2 groups? Since patients have to pay out of pocket for 50% of the FDC, how many patients agreed to the switch and persisted with the new medications? On this point, did patients have access to a list of free medications for BP control? At baseline, the patients were already on multiple drugs and what was the pattern of drug usage between the 2 groups at baseline and end of 24 months? 

In the intervention group, were the patients informed of their BP goals and were they taught to self monitor BP? Indeed having a coordinator to relay these 2 messages every 6 months or so might go a long way while allowing the doctors and patients make their own choices in term of FU visits, drug regimens and lab tests. Indeed, this may be the reason why both groups had dropped their BP significantly with the yearly review by the project coordinator.

The authors should list the names of all 8 participating clinics and their PIs in the acknowledgement. 

Reviewer #3: Alex McConnachie, Statistical Review

Jafar et al present the results of a cluster randomised trial of a multicomponent intervention in primary care to improve blood pressure control. This review considers the statistical aspects of the paper.

Overall, this is an excellent study, very well reported. The statistics are generally very good, and my comments are therefore relatively minor.

One difficulty with cluster randomised trials is always the timing of randomisation at the cluster level and recruitment of individual patients. My impression from the paper is that practices were randomised first, and then the patients were recruited. What is not clear is whether the patients were aware of the allocation of their practice at the time of giving consent to be included in the study, or when they provided their baseline data. Also, how long was it between randomisation of the practices and recruitment/baseline? What was the timing of the intervention - had any training taken place before the patients had completed their baseline assessments? Were the follow-ups conducted 12- and 24-months after each patient's baseline assessment, or relative to the randomisation of the practice? Line 228 refers to the 2-year follow-up post-randomisation, but was it actually 2 years post-baseline?

Line 231 refers to blood pressure <140/90 as an outcome, though there was an earlier reference (line 159) to <130/80 as a target for some patients. Was this lower target used for relevant patients in defining this outcome?

The sample size calculation appears correct, but it is not possible to replicate exactly without knowing what loss to follow-up was assumed.

The repeated measures approach to the analysis is good, and is described very well. The sensitivity analyses are all very sensible, though the definition of "per protocol" is not clear - I did find it in the footnote to one of the tables, but it could perhaps be included in the main text. In sensitivity analysis 4, I assume the covariates were baseline measures?

Some people would complain about reporting p-values for baseline differences, but I think it is a good thing with cluster randomised trials, and it is good that these comparisons take account of the clustering.

In Tables 2 and 3, we are given the mean and 95% CI, or percentage with 95% CI, in each randomised arm. This is not wrong, but for me, it does not show the distribution of the data in each group, since the CI depends on the sample size. Personally, I would rather see a summary of the data, i.e. mean and SD, or number and percentage, within each group.

Reporting ICC statistics (line 377) is good, though it would be good if these could be reported for all outcomes, perhaps within the supplement somewhere.

Finally, in Figure 1, the variance in cluster sizes in the two arms are reported as 21 and 13 "per cluster". I'm not sure that "per cluster" makes sense, since the units of variance is the square of the units of measurement. Would it be easier to report the SD of cluster sizes?

Reviewer #4: MS: PMEDICINE-D-22-00512R1

Summary:

This study investigates the effects of multicomponent intervention for managing hypertension. In this cluster randomized controlled trial (RCT) within a primary care setting in Singapore, an intervention that integrates CVD risk assessment, subsidised single pill combination drugs, motivational conversations and telephone follow-up was compared to standard care in lowering blood pressure (BP). After a follow-up of up to two years, the investigators report a reduction of 3 mmHg systolic BP and better control of hypertension in the active arm compared to standard care. 

Comments:

This trial is indeed interesting in that factors beyond that of the drug therapy itself is being considered and reflect effectiveness of an intervention that are relevant to practice settings. The inclusion of CVD risk assessment is an important strength, as treatment benefit tends to be higher in people with a higher background risk, at least in absolute terms. However, an important limitation is that it recruited participants with baseline BP levels of 140/90 mmHg or higher. I would have thought that CVD risk assessment would have benefited those at increased risk but their baseline BP may be slightly below this BP thresholds (e.g., hypertension treatment may be initiated in patients with diabetes whose BP are below this threshold as some clinical guidelines would suggest). 

The achieved BP difference (of just over 3 mmHg systolic) seems very modest - clinically important at a group level, but likely less so at individual level. Because the aim of the intervention is to control hypertension or bring BP levels below the 140/90 mmHg threshold, which is the same for both treatment arms, it seems that the added benefit of all the components only amounted to 3 mmHg of effectiveness - a huge undertaking for a modest impact. Rather than focusing on the various components, would it have been better to focus on a component of the intervention that is most likely to achieved a greater reduction, perhaps by ensuring reaching an optimal target that is well below 140/90 mmHg? Very few (<6%) were at 'low-risk' of CVD at baseline, and the vast majority were either at medium or high risk. What would have been the impact if everyone was simply offered single pill combination drugs - it would have simplified further the intervention, and even if the costs might be slightly higher, the benefit in reducing BP perhaps might be freater?

The inclusion criteria included BP levels of ≥140/≥90 mmHg based on mean of last 2 of 3 measurements. Since a good proportion of participants were on antihypertensive medications at baseline, does it mean that either these patients had poor BP control and/or had higher baseline BP if they were not on medications? 

Making the most of BP measurements in reporting BP differences is indeed more appropriate than simply comparing BP changes from baseline. However, it is expected that BP measured a few months after randomisation tend not to reflect the full effects of the intervention since several adjustments/titration take place during this period, as well as regression to the mean/random variability in BP. I suggest that the authors consider a sensitivity analysis that exclude BP measured, say, in the first 3 or 6 months - it might be possible to demonstrate a bigger treatment effect than the 3 mmHg systolic BP being reported.

The difference in the albumin:creatinine ratio seems very substantive which is not in keeping with differences in other parameters between treatment arms. Was this an expected finding? Is this a plausible result given the small difference in achieved BP reduction between the comparison groups?

[LINK]

---

## [Decision Letter · Decision Letter 2]

10 May 2022

Dear Dr. Jafar,

Thank you very much for re-submitting your manuscript "Integration of a Multicomponent Intervention for Hypertension into Primary Healthcare Services - A Cluster Randomised Controlled Trial" (PMEDICINE-D-22-00512R2) for review by PLOS Medicine.

I have discussed the paper with my colleagues and the academic editor and it was also seen again by four reviewers. I am pleased to say that provided the remaining editorial and production issues are dealt with we are planning to accept the paper for publication in the journal.

[LINK]

We look forward to receiving the revised manuscript by May 17 2022 11:59PM.   

Sincerely,

Callam Davidson, 

Associate Editor 

PLOS Medicine

plosmedicine.org

Requests from Editors:

Please update your title to ‘Integration of a Multicomponent Intervention for Hypertension into Primary Healthcare Services in Singapore - A Cluster Randomised Controlled Trial’.

The Data Availability Statement (DAS) requires revision. If the data are not freely available, please describe briefly the ethical restriction that prevents you from sharing it. 

Please note that line numbers in the comments below pertain to the tracked changes word document you provided.

Line 169: Please cite your protocol using our Supporting Information guidelines (e.g., S1 protocol, or similar). 

Guidance can be found here: https://journals.plos.org/plosmedicine/s/supporting-information

Line 285: This sentence can now be removed as you already state this on line 256.

Table 1 and Line 409: With apologies for the conflicting advice I gave on the previous version, as noted by the statistical reviewer, in this particular case it is OK to present p-values for a comparison of baseline data (see comments from Reviewer 3 for more information).

Lines 688-690: Please remove funding information from the Acknowledgements.

Reference 8 contains COI information that can be removed.

Supporting Information: Please include references to sections S1-4 in the main text.

Figure 3 and Table 3: The terms gender and sex are not interchangeable (as discussed in http://www.who.int/gender/whatisgender/en/ ); please use the appropriate term. 

Comments from Reviewers:

Reviewer #1: no other comments.

Reviewer #2: The revised paper is considerably improved and the authors have addressed all the concerns satisfactorily. A few points need clarification:

1. Line 172-173: Generic and branded SPC antihypertensive medications are also available, albeit they are not subsidized - are generic medications also NOT subsidized?

2. Ref 1 and 2 need correction 

3. Ref 8 is mixed with declaration on conflicts of interest

Reviewer #3: Alex McConnachie, Statistical Review

Revision of "Integration of a Multicomponent Intervention for Hypertension into Primary Healthcare Services - A Cluster Randomised Controlled Trial", Jafar et al.

I thank the authors for their consideration of my original points.

Whilst I appreciate that the paper is clear that the study was not blinded, I think it is worth stating explicitly that the patients were aware of the group that their practice had been assigned to at the point of enrollment. This should also be recognised as a limitation.

This has implications for the presentation of the baseline data. As the editors say, for randomised trials it is not usually correct to present p-values for a comparison of baseline data. However, the fact that patients were aware of their allocated group at baseline data collection means that there could be systematic differences between the patients recruited in the two arms. We need reassurance in the form of statistical comparisons that this is not the case.

In Tables 2 and 3, I don't think adding the standard errors improves things (this is essentially the same information as given by the CIs). I am aware that the mean values reported are model predictions, which was my point; I would rather see a summary (mean and SD, or N and %) of the actual data in each group at each time point. Reporting model-predicted values and CIs is fine in the figures, and of course, we need the estimated between-group differences and ORs, but I feel it is important to present some summaries of the data that was actually collected.

Reviewer #4: I have no further comments.

[LINK]

---

## [Decision Letter · Decision Letter 3]

20 May 2022

Dear Dr Jafar, 

On behalf of my colleagues and the Academic Editor, Dr David Peiris, I am pleased to inform you that we have agreed to publish your manuscript "Integration of a Multicomponent Intervention for Hypertension into Primary Healthcare Services in Singapore - A Cluster Randomised Controlled Trial" (PMEDICINE-D-22-00512R3) in PLOS Medicine.

PRESS

Sincerely, 

Callam Davidson 

Associate Editor 

PLOS Medicine